# Effect of Abiotic Climatic Factors on the Gonadal Maturation of the Biocontrol Agent *Sphaerophoria rueppellii* (Wiedemann, 1830) (Diptera: Syrphidae)

**DOI:** 10.3390/insects13070573

**Published:** 2022-06-24

**Authors:** José J. Orengo-Green, José L. Casas, Mª. Ángeles Marcos-García

**Affiliations:** Unidad Asociada (UA-CSIC) IPAB, Research Institute CIBIO (Centro Iberoamericano de la Biodiversidad), Science Park, University of Alicante, Ctra, San Vicente del Raspeig s/n, 03690 Alicante, Spain; jl.casas@ua.es (J.L.C.); marcos@ua.es (Mª.Á.M.-G.)

**Keywords:** Gonad’s description, lifetime fecundity, oviposition, photoperiod, predatory hoverfly, temperature

## Abstract

**Simple Summary:**

Knowledge about the morphology and functioning of the male and female reproductive system in insects is key to understanding their reproductive biology, and to assessing the effects that environmental factors, such as temperature or photoperiod, can have on oviposition, fecundity, and lifespan. This knowledge is particularly interesting in those species that are mass-reared, as in the case of the predatory syrphid *Sphaerophoria rueppellii*. Given the lack of published information regarding sexual maturation in syrphids, this type of study, applied to beneficial insects used as biological control agents, offers, firstly, the chance to improve their mass breeding under controlled conditions and, secondly, to know their capability for pest control response under field conditions. Our results show that photoperiod and temperature affect development and gonad maturation in *S. rueppellii* males and females.

**Abstract:**

The hoverfly *Sphaerophoria rueppellii* is currently one of the most effective predators commercially available for aphid pest control. However, knowledge of the reproductive system of males and females of this syrphid is limited. The present article aims to report how changes in the temperature and photoperiod may affect development of the gonads (ovaries and testes), oviposition, and fecundity during the lifespan of *S. rueppellii.* Four environmental conditions (14L:10D, T: 20 ± 1 °C; 12L:12D, T: 20 ± 1 °C; 14L:10D, T: 25 ± 1 °C; and 12L:12D, T: 25 ± 1 °C) were used to determine oviposition, hatching percentage, and lifespan during a period of 30 days after the adult emergence. The maturation of the ovaries was done under three treatments (barley leaves with aphids always available; barley leaves two days per week with aphids available; no barley leaves available), and in the same environmental conditions noted above. Males at 14L:10D, 20 ± 1 °C; and 14L:10D, 25 ± 1 °C; were used to analyze and study the maturation of the testes. Females at 14L:10D; T: 25 ± 1 °C showed a significant difference in oviposition, percentage of hatching, and rate of eggs. A detailed description of the male and female gonads was undertaken, and it was determined that the conditions in which males sexually mature early are at 14L:10D, 25 ± 1 °C. These results will improve the application of *S. rueppellii* in crops, for the control of aphid pests.

## 1. Introduction

The Syrphidae family has more than 6000 species, highly distributed worldwide [1,2,3]. Adult syrphids feed on nectar as a source of energy and pollen, to mature sexually and complete their reproductive cycle [3], which makes them good pollinators [4,5,6,7]. Syrphids are considered to be efficient aphidophagous agents in biological control [5,8,9,10,11], because a third of syrphid species larvae prey on small and soft Hemiptera, aphids being their main prey [12]. Aphids are among the most serious agricultural pests [13], due to plant virus transmission [14], direct plant damage [15], and insecticide resistance [16,17]. The use of biological controls, using parasitoids and aphid predators, is becoming increasingly necessary and promising for the control of pests [11,18,19]. For this reason, three predatory syrphid species are currently commercially available: *Episyrphus balteatus* (De Geer, 1776), *Sphaerophoria rueppellii* (Wiedemann, 1830), and *Eupeodes corollae* (Fabricius, 1794) [11,20,21].

Surprisingly, very few studies concerning the reproductive system and oviposition of aphidophagous syrphids have been conducted up to now. An exception is *E. balteatus*, a common Palearctic hoverfly, of great importance as a natural enemy [6,18,22,23,24]. In this species, adult feeding affects longevity [25] and oviposition [24,26]. In addition, the presence of oviposition stimuli directly affects the maturation of the ovaries, the number of eggs produced, and the absorption of the eggs [22,23,27], but nothing is known about the sexual maturation of the males. A study made on potential biological control agents [28] revealed that low temperatures negatively affect the oviposition of the Nearctic syrphid *Eupeodes americanus* (Wiedemann, 1830), as they affect the number of eggs laid. There are detailed morphological descriptions of the reproductive systems of saprophagous syrphids species belonging to different phylogenetic groups, such as *Eristalinus taeniops* (Wiedemann, 1818), *Eristalinus aeneus* (Scopoli, 1763), and *Eristalis tenax* (Linnaeus, 1758) [29]. However, there are no studies carried out on morphology, or on the influence of abiotic climatic factors, gonadal maturation, and lifetime oviposition of the syrphid predator *S. rueppellii*.

In recent years, *S. rueppellii* has been used in some European countries as a biocontrol agent against aphid pests [11]. This Palearctic species can be found easily, in its natural surroundings of crops—both fruit trees and horticultural—and inside greenhouses, in the Mediterranean area, being more abundant in Southern Europe and North Africa [30,31,32]. As in other holometabolic insects, the life cycle of this species depends mainly on humidity, photoperiod, and temperature, in which it has a different range of optimal development [20]. The life cycle of *S. rueppellii*, from egg to adult emergence at 25 ° C and 90% RH, is about 17 days [20]. Syrphids females are not born sexually mature, because they need to feed on pollen to be able to complete the maturation of their ovaries [18,20,22,33]. In the case of *E. balteatus*, it has been shown that the development time from larva to adult depends on photoperiod and temperature: decreasing the number of light hours and temperature slows down the cycle development time, but the adults live longer [34,35]. However, little is known about the effects of photoperiod and temperature on the reproductive systems of the females and males of *S. rueppellii*.

The aim of this study is therefore to increase knowledge of the reproductive biology of the predatory syrphid *S. rueppellii* under different conditions of light and temperature, these two variables being the main factors that trigger the diapause or gonadal development process in these diptera [35]. We aim to answer the following questions: (1) what is the oviposition lifetime at different conditions of temperature and photoperiod; (2) does ovarian resorption occur in *S. rueppellii* as in other aphidophagous syrphid species; (3) do temperature and photoperiod affect the development and maturation of the ovaries and testes of *S. rueppellii*?

The results for the knowledge of the reproductive biology of *S. rueppellii* may help to improve the conditions for its management during mass rearing, and its application in crops for the control of aphid pests. Fundamental knowledge of the reproductive biology of natural enemies can help to design more effective Biological Control strategies and Integrated Pest Management programs [36].

## 2. Materials and Methods

### 2.1. Experimental Insects and Morphological Terminology

The pupae of *S. rueppellii* used in all trials were provided by BioNostrum Pest Control S.A. (Alicante, Spain). The syrphid pupae were placed in open Petri dishes inside rearing cages with fine gauze walls (43 × 43 × 37 cm). After emergence, adults were maintained with bee granular pollen, a solution of water, and commercial white sugar presented on a 100 mL plastic glass inverted on a Petri dish lid with a disc of filter paper replaced every two days. The environmental conditions were performed in a Fitoclima 10,000 EHVP phytotron chamber.

The morphological terminology used for the ovaries follow Branquart and Hemptinne [22], and for the male internal reproductive organs it follows Sinclair et al. [37]. Egg load (= rate of egg) is defined as the number of mature oocytes available for oviposition [22,38].

### 2.2. Lifetime Oviposition

The daily egg-laying was determined by following a total of 600 females of *S. rueppellii* for a minimum of 6 days to a maximum of 30 days after their emergence (according to own preliminary test). Some trials were done to determine the beginning of the oviposition day, based on the work done by Amorós-Jiménez et al. [21]. Four environmental conditions, combining temperature and photoperiod, were tested (Table 1); for each of them, 400 pupae were used, to ensure the number of females required for the assays. After emergence, the adults were distributed in five fine gauze containers (43 × 43 × 37 cm), each containing 30 females and 10 males to ensure copulation [22,39]. In all assays, the oviposition site was four freshly cut barley leaves of about 20 cm long, with 50 wingless adults of the aphid *Rhopalosiphum padi* (Linnaeus, 1758). Each of the leaves was placed daily for 30 days into the container to stimulate egg laying. After 24 h. all the leaves were removed, and the deposited eggs were counted under a Leica M205C stereomicroscope. Additionally, the mean oviposition lifespan and percentage of egg hatching were calculated. Because hatching of the egg has been described as occurring three days after the laying [20], the percentage of fertility was recorded by counting daily all the hatched larvae at the same hour.

### 2.3. Ovaries Maturation

To test the effect of environmental conditions on ovaries maturation in females of *S. rueppellii*, 1200 pupae were distributed and maintained under the conditions described in Table 1. After the emergence of the adults, groups of 30 females and 10 males were distributed in gauze containers (43 × 43 × 37 cm), and further subdivided into three layouts: Treatment I: containers with four barley leaves with aphids throughout the complete experiment; Treatment II: container with aphids on four barley leaves only two days a week; Treatment III: containers without barley leaves. Five containers were used for each of these treatments; therefore, a total of 1800 females of *S. rueppellii* were analyzed for ovary maturation during the 30 days following their emergence. To assess ovary maturation, five females from each treatment (one from each container) were randomly captured every three days; their abdomens were carefully dissected using micro scissors, following the dissection methodology previously described by Gilbert [40] (Figure 1), and the mature ovarioles were counted under the stereomicroscope. From emergence, the adult syrphid were provided with water, pollen, sugar, and oviposition stimuli (for details, see Section 2.1). The ovaries were carefully removed and placed in glycerin for examination under a Leica M205C binocular stereomicroscope. Photos were produced as a stack of individual images produced by a camera (Leica DFC 450, Leica Camera AG, Wetzlar, Germany) attached to a binocular stereomicroscope (Leica M205C). Stacks were made in the Leica Application Suite LAS^®^, v.4.12.0., Leica Microsystems, Wetzlar, Germany.

### 2.4. Maturation of the Testes

To assess when males of *S. rueppellii* acquire their sexual maturity, and the potential influence of environmental conditions, a total of 800 pupae were used to guarantee the number of specimens required for the whole experiment. When adults started emerging, males were separated to precisely control their age, and thus to obtain males from 1 to 3 days old. Ten males of each age were placed along with 30 mature females in gauze containers, as described previously. Six containers were used for each of the environmental conditions indicated in Table 1. The barley leaves included in the containers were recorded daily, to detect the first day of oviposition and hatching. For 8 days, a randomly selected male from each container was dissected, to observe the testes maturation. The testes were dyed in fuchsine, to facilitate the observation of the transparent organs, and observed and photographed as described in the above section. This assay was made under the following conditions: 20 °C and 25 °C, both with a photoperiod of 14L/10D, due to this being the one with the best results in the *Lifetime fecundity and ovaries maturation* section.

### 2.5. Statistical Analysis

Data analyses were carried out for comparison between various parameters (lifetime, oviposition, treatment, percentage of hatching, and egg load), using a Tukey test at a 5% significance level. Napierian logarithm was used for the Tukey test, on the analysis of the number of eggs laid. All the statistical analyses were carried out with the Statistical Program InfoStat Ver. 2018e, Universidad Nacional de Córdoba, Argentina [41].

## 3. Results

### 3.1. Lifetime Fecundity, Fertility, and Lifespan

The egg laying started on the 7th day after the emergence, increasing daily until reaching the maximum point, and then slowly decreasing but maintaining the oviposition until the death of the females (Figure 2). The oviposition pattern was found to be influenced by the environmental condition assayed. Thus, under A or B conditions the highest oviposition appeared on the 10th day; however, increasing the temperature to 25 ± 1 °C led to a delay in the maximum period of oviposition, of 1 (condition D) and 2 days (condition C), according to the photoperiod (Figure 2). The mean oviposition during the whole female lifespan also varied according to the environmental condition, with condition C being the highest (294 ± 28.35), and condition D the lowest (94.05 ± 30.07) (Table 2). A significant difference was found between C and the rest of the conditions (*p* < 0.0001). Under a low temperature (A and B conditions), though the mean of eggs/females was lower, the lifespan oviposition lasted for a week longer than at a higher temperature (C and D conditions) (Figure 2). The mean percentage of hatching was better at condition C (96 ± 1.38), and the lowest was at condition D (81 ± 0.73) (Table 2). Condition C shows a significant difference from the rest of the conditions (*p* < 0.0001) (Table 2).

A significant difference was found in the lifespan of females and males between the environmental conditions applied (Table 3). The longest lifespan was found under conditions A for males (28 ± 2.3 days) and B for females (29.6 ± 0.92 days). However, the shortest lifespan was condition C for males (20.8 ± 0.86 days) and D for females (20.7 ± 1.24 days). Male and female longevity, without considering the environmental condition, was 23.4 ± 0.93 and 25 ± 0.93 days on average, respectively, and it was not significantly different. These results show that the photoperiod and temperature influence the fecundity and longevity of the adults of *S. rueppellii*.

### 3.2. Ovaries Maturation

During the maturation week following the emergence, the number of oocytes in the ovarioles successively increased until they reached maturity, when an oocyte was fully developed in each ovariole (Figure 3). At this moment, the maximum number of oocytes in formation per ovariole was five (plus the apical mature oocyte), and a total of 21 ovarioles per ovary. The ovariole was meroistic polytrophic, with a nurse cell and a cystomatic bridge observable (Figure 3). Both the maturation day (when the first mature oocyte appears) and the day of reaching the maximum number of ovarioles were affected by the different situations (Table 4). The egg load per ovary was influenced (*p* < 0.0001) by the conditions assayed (Figure 4 and Figure 5). *S. rueppellii* females produced one to two eggs per ovariole per day in Treatments I and II, but in Treatment III (without barley leaves) the rate was lower than one egg. There was a significant difference in the rate of eggs between Treatments (Figure 5). Interestingly, in Treatment III, the females laid infertile eggs in the vicinity of the water and food container. This could be confirmed because we waited longer than the time estimated for the hatching of the eggs [20]. Egg resorption in the ovaries was not seen under any of the conditions and treatments here assayed.

### 3.3. Description of the Male Reproductive System of Sphaerophoria rueppellii

The male reproductive system is relatively easy to identify under a binocular stereomicroscope, due to the yellowish coloration of the epithelium. It is formed by two independent oval testes (T) apically narrowed and filled with sperm. They are located in the apical end of the abdomen, extending from the fourth to the ninth abdominal segments (Figure 6A). The two vas deferens (VD) are very short, and they are inserted on the basal part of the testes at its junction with the ejaculatory tract (ET) (Figure 6A). The ET is the same color as the testes, and it has a rough surface due to the fatty bodies. Inside the ET, two distinguishable filaments of sperm can be observed (Figure 6B). At the end of the ET, two accessory glands (AG), resembling large, translucent whitish tubes, and the seminal glands (SV), can be observed (Figure 6A). A thin tracheal filament (TF) accompanies the entire reproductive system (deleted in the image).

### 3.4. Testes Maturation

One-day-old males at 25 °C copulated with mature females, but these eggs were infertile. Most of the males held at 25 °C copulated at 2 days old, and the females laid fertile eggs the next day. When the temperature decreased, it was observed that the male maturation slowed down. One-day-old males (*n* = 60) at 20 °C started copulating on the 5th day after emergence, but the eggs laid were infertile. Males at 20 °C copulated by the 6th day after emergence, resulting in the laying of fertile eggs. At 20 °C, it took 5 days to develop a mature testicle. When dissecting mature males, the testes were ventrally observed to extend from the fourth to the ninth abdominal tergites, but the size of the testes was smaller on immature males, occupying from the fifth to the ninth abdominal tergite.

## 4. Discussion

Our results indicate that *S. rueppellii* has a lower number of ovarioles (42) but a higher egg load (210) at temperatures that are typical for greenhouse crops. The female of *S. rueppellii* produces one or two eggs per ovarioles per day, which coincides with data provided by [22] for *E. balteatus*. The aphidophagous syrphid *E. balteatus*, whose adults are about twice the size of *S. rueppellii*, has 80 ovarioles and an egg load of approximately 130 [22]. It must be considered that a variation from the optimum conditions may lead to a potential change or delay in oviposition, maturity, rate of egg, and lifespan. For this reason, the number of ovarioles is an important asset to determine the fecundity of a species [42,43] and, therefore, its potential as a biological control agent. Our data concluded that the number of fertile eggs laid per day by *S. rueppellii* mature females increased with the presence of an oviposition stimulus. When an oviposition site was available, females began to lay eggs daily when the ovaries contained mature oocytes, and continued to do so until the death of the female or the stimulus was removed [22]. This agrees with Jervis et al. [44] who noted that insects keep producing eggs as they age, but that the egg production decreases. This is similar to what we observed in our results. In addition, it has been seen that by having an oviposition site present, females lay eggs more frequently [45,46].

The lifetime oviposition during the whole lifespan of *S. rueppellii* depended on temperature and photoperiod, varying from 20 to 30 days (Figure 3). This value was in the range of 25 days for *E. corollae* [47] and 30 days for *Pseudodorus clavatus* (Fabricius, 1794) [48], but was lower than the 50 days for *E. balteatus* [22]. A possible explanation for the higher values of *E. balteatus* compared to the other three aphidophagous syrphid species may be its larger body size, and in particular its greater abdominal capacity, a characteristic that in syrphids has been considered an indicator of the species’ fitness, as it allows for the hosting of a larger number of ovarioles [21,49]. In our study, we observed that the peak of oviposition (Figure 2) was delayed by 1–2 days by increasing the temperature from 20 to 25 °C, but that under lower temperature, the oviposition lasted for a week longer. This is important in the establishment of biological control preventive strategies, such as when pupae should be released in the field for adult emergence. Another result to consider is that the higher temperature and longer photoperiod increased the number of eggs laid by *S. rueppellii* (Table 2). Similar results can be seen in the work done by Hondelmann and Poehling [35] on *E. balteatus,* which notes that in lower photoperiod time (11 L, 10 L, and 8 L), the development of the ovaries ceased almost completely, leading to the conclusion that the egg laying decreased in the lower photoperiod but increased in the longer photoperiod time. Other insects, such as Hymenoptera parasitoid, parasite (=egg laying) more in intermediate temperatures (20–30 °C) than in margin temperatures (15 °C and 35 °C) [50].

In our results, *S. rueppellii*, in the absence of aphids as oviposition stimulus, did not show any symptoms of eggs resorption; females, however, laid unfertilized eggs around different substrates and locations. This implied that to lay fertile eggs they needed the presence of prey, this being a wanted behavior for an efficient natural enemy. The presence of available aphids affects the fecundity of the females, as has been previously noted, and its absence may result in the resorption of eggs observed in *E. balteatus* and *E. corollae* [22,51]. In our results, *S. rueppellii*, in the absence of aphids as oviposition stimulus, did not show any symptoms of egg resorption, but the females laid unfertilized eggs. A possible explanation for the ovarian resorption in *E. balteatus* and *E. corollae* is that they are migratory species [35,52] looking for new plants with aphid colonies, and to complete this distance they used the energy of un-laid eggs as a survival tactic to avoid wasting energy [44,53,54]. As no egg resorption was found in *S. rueppellii*, the egg numbers were limited by the storage capacity of the ovariole, meaning that the ovigenesis could cease, as there was no more room for eggs [44].

The composition of the ovarioles of *S. Rueppellii* is like that of *E. balteatus*, as both are composed of six oocytes per ovariole [22]. The type of ovariole present in our results was consistent with previous studies [44,55], which noted that the Diptera species have meriostic polytrophic ovarioles. The presence of nurse cell and cystomatic bridges are a more effective way to provide the necessary nutrients for the development of the oocyte. Having identified these two parts, it was possible to confirm that it is a type of ovariole present in most holometabolous insects [56].

The longevity of the adults of *S. rueppellii* varies with the temperature; both males and females and lay eggs for one more week at lower temperature (20 °C). This result agrees with Hong and Hung [57], who demonstrated the influence of the temperature on the lifespan of the syrphid *E. balteatus*. In some syrphids species, it has been observed that the female lives longer (2 days more) than the male [25], and this was confirmed by our results. The females of *S. rueppellii* live (25 ± 0.93 days) compared to males (23.4 ± 093 days), although no significant differences were found. This result agrees with van Rijn et al. [27], who found no significant difference in lifespan between the sexes of *E. balteatus*. A possible explanation for the difference in longevity between sexes may be that mating causes a positive outcome in the lifespan of female syrphids, possibly due to the greater energetic effort of the males to ensure copulation [45]. Similar results can be seen in the parasitoid *Diaphorencyrtus aligarhensis* (Shafee, Alam, and Argawal) (Hymenoptera: Encyrtidae) that lives longer at lower temperatures than at higher temperatures [58].

To the best of our knowledge, no work has been done on aphidophagous syrphids, in regard to the study of the male gonads and their maturation time. The morphology of the male genital tract in Diptera is poorly known, with only a few detailed comparisons having phylogenetic significance between some families [37]. In most insects, the male reproductive system is composed of a pair of testes connected to a seminal vesicle by the vas deferens, accessory glands, and an ejaculatory tract [59]—all these components being present in *S. rueppellii*. However, the length of the vas deferens of *S. rueppellii* is noticeably shorter than in other Diptera families such as Tipulidae or Psychodidae [37,59]. The accessory glands in *S. rueppellii* are located posteriorly to the ejaculatory tract (Figure 6A), and are morphologically similar to those of other Diptera. The function of the accessory glands is to synthesize proteins that are transferred with the sperm to the female during mating [59]. According to [40], syrphids males need to have mature testes and accessory glands to be able to fertilize the female. In *S. rueppellii*, there is no appreciable morphological difference between mature and immature accessory glands.

Our results are important, because they broaden the knowledge about how the two main climatic conditions affect the development and reproduction of this syrphid species that is used as biological control. These results are especially important because temperature is one of the more variable climatic factors. Moreover, the new data, on the reproductive system of male syrphids, open a new path for the study of the morphology, biology, and behavior of the syrphid.

## Figures and Tables

**Figure 1 insects-13-00573-f001:**
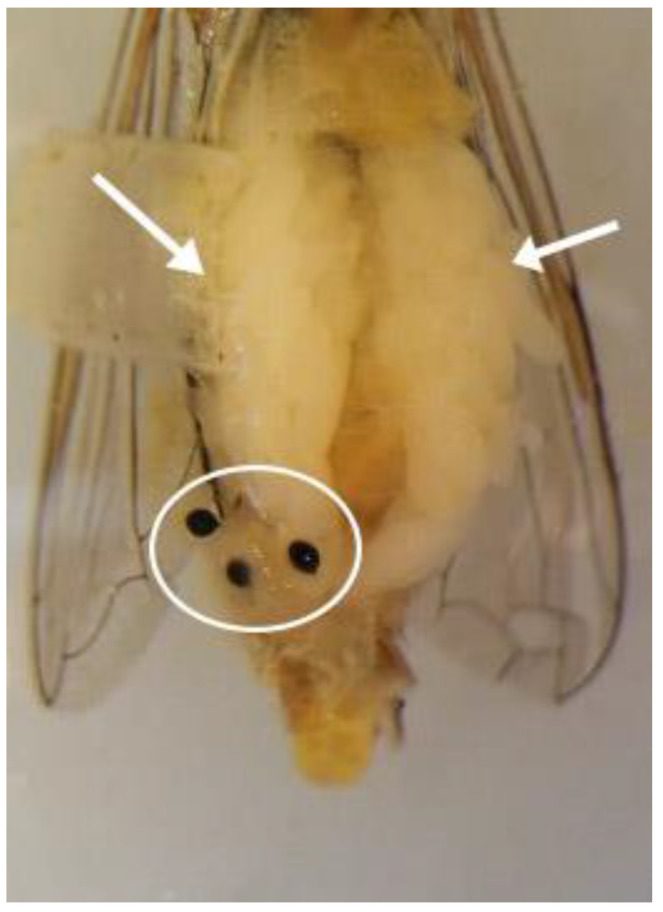
Dissected female of *Sphaerophoria rueppellii* showing the ovaries (arrows) and three spermatheca (circle).

**Figure 2 insects-13-00573-f002:**
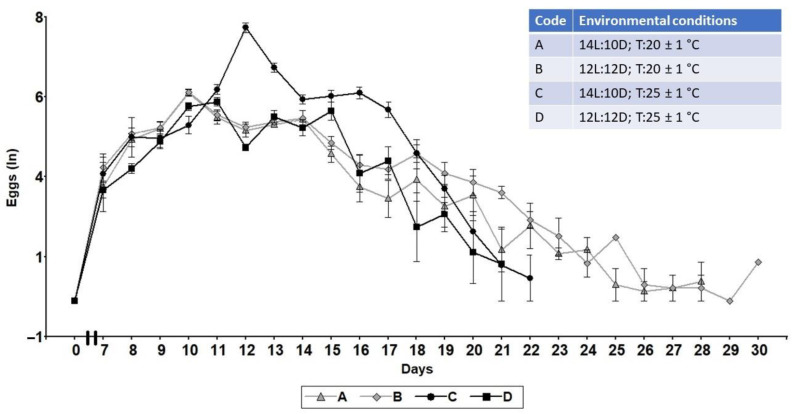
Lifetime oviposition of *Sphaerophoria rueppellii* (*n* = 150) during the whole lifespan under the four different environmental conditions studied.

**Figure 3 insects-13-00573-f003:**
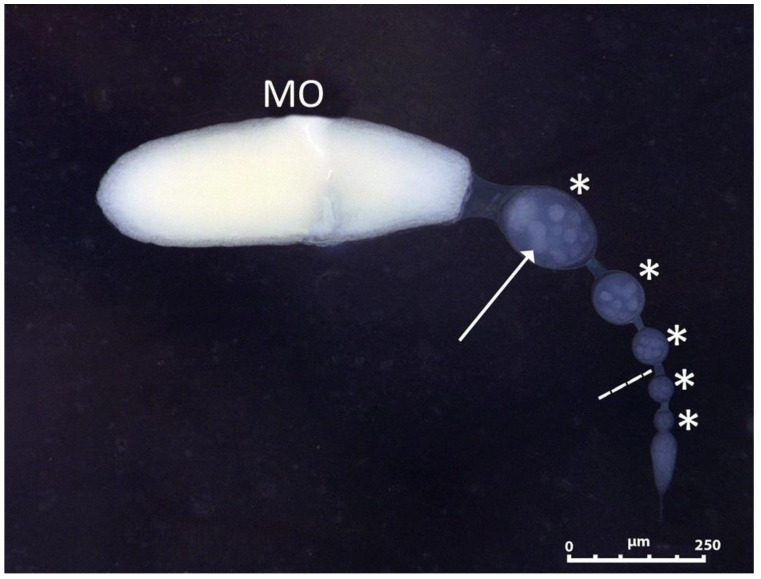
Schematic representation of a mature meroistic polytrophic ovariole (MO: mature oocyte; asterisk: immature oocyte; arrow indicates nurse cells; broken line indicates cystomatic bridge).

**Figure 4 insects-13-00573-f004:**
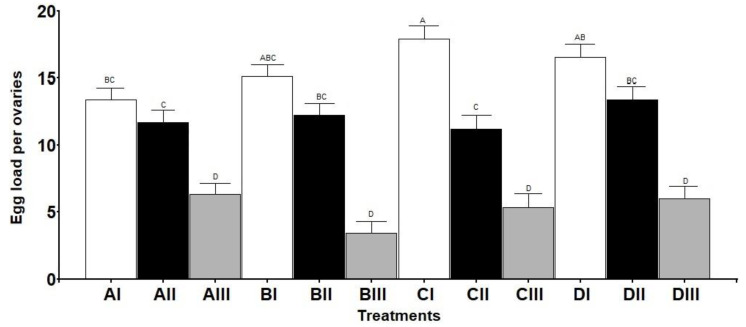
Egg load per ovary of *Sphaerophoria rueppellii*. Data are the mean *±* SE of egg load per ovary (*n* = 150) during a period of 30 days. Equal letters are not significantly different according to Tukey’s test (*p* ≥ 0.05).

**Figure 5 insects-13-00573-f005:**
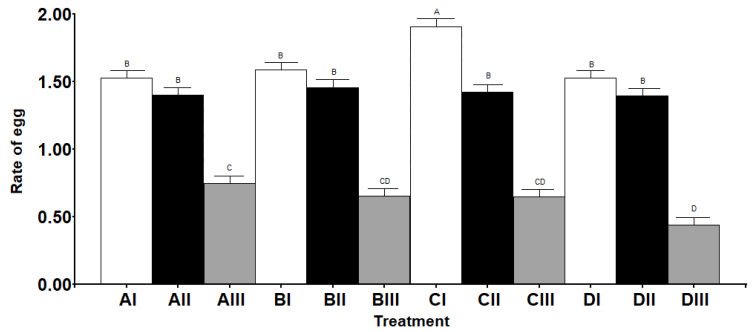
Rate of egg production of *Sphaerophoria rueppellii*. Data are the mean ± SE rate of eggs at the peak of ovarioles. Equal letters are not significantly different according to Tukey’s test (*p* ≥ 0.05) (*n* = 60).

**Figure 6 insects-13-00573-f006:**
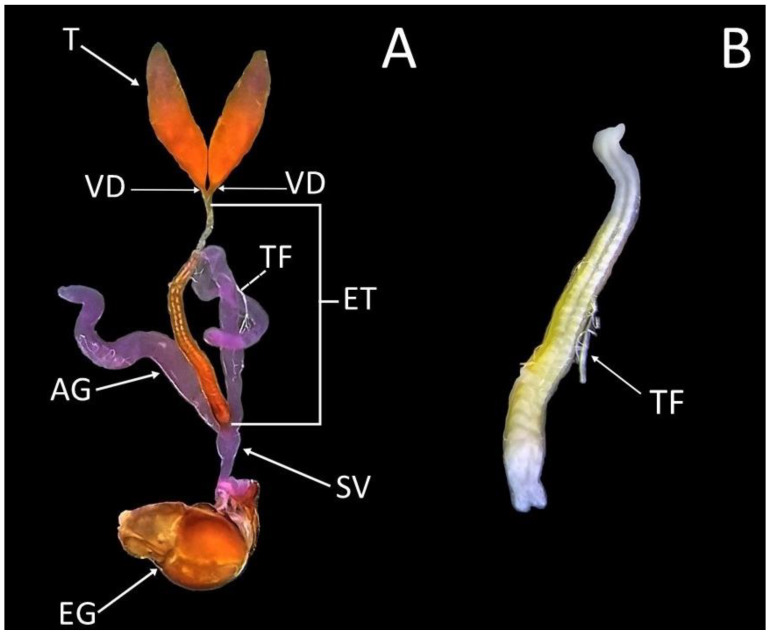
*Sphaerophoria rueppellii* male reproductive organs: (**A**) Dorsal view (AG—accessory glands; EG—external genitalia; ET—ejaculatory tract; SV—seminal vesicle; T—testes; TF—tracheal filament; VD—vas deferens); (**B**) Ejaculatory tract.

**Table 1 insects-13-00573-t001:** Environmental conditions assayed on each of the four assays. In all cases, HR was always 85 ± 5%.

Code	Environmental Conditions
A	14L:10D; T:20 ± 1 °C
B	12L:12D; T:20 ± 1 °C
C	14L:10D; T:25 ± 1 °C
D	12L:12D; T:25 ± 1 °C

**Table 2 insects-13-00573-t002:** Mean of oviposition per female of the whole lifespan (*n* = 150). Mean ± SE with equal letters meant no significant difference according to Tukey’s test (*p* ≥ 0.05).

Code	Environmental Conditions	Mean Total Eggs	Hatching (%)
A	14L:10D, 20 ± 1 °C	62.72 ± 24.06 ^b^	88.6 ± 0.87 ^a^
B	12L:10D, 20 ± 1 °C	70.91 ± 24.06 ^b^	91 ± 0.55 ^a^
C	14L:10D, 25 ± 1 °C	294.67 ± 28.35 ^a^	96 ± 1.38 ^b^
D	12L:12D, 25 ± 1 °C	94.05 ± 30.07 ^b^	81 ± 0.73 ^a^

**Table 3 insects-13-00573-t003:** Lifespan of *Sphaerophoria rueppellii* under different environmental conditions. Mean *±* SE with equal letter within columns is not significantly different according to Tukey’s test (*p* ≥ 0.05).

Code	Environmental Conditions	Lifespan (Days)	
		Males (*n* = 50)	Female (*n* = 150)
A	14L:10D, 20 ± 1 °C	28 ± 2.3 ^a,b^	27.1 ± 0.72 ^a,b^
B	12L:10D, 20 ± 1 °C	23 ± 0.89 ^b,c^	29.6 ± 0.92 ^a^
C	14L:10D, 25 ± 1 °C	20.8 ± 0.86 ^c^	22.4 ± 1.25 ^c^
D	12L:12D, 25 ± 1 °C	21.8 ± 0.80 ^c^	20.7 ± 1.24 ^c^

**Table 4 insects-13-00573-t004:** Days to the first mature oocyte and peak of ovarioles.

Code	Environmental Conditions	Treatment *	Maturation (Day)	Peak of Ovarioles (Day)
A	14L:10D, 20 ± 1 °C	I	8th	11th
		II	11th	15th
		III	11th	15th
B	12L:10D,20 ± 1 °C	I	8th	11th
		II	8th	11th
		III	8th	11th
C	14L:10D, 25 ± 1 °C	I	8th	11th
		II	8th	11th
		III	8th	15th
D	12L:12D, 25 ± 1 °C	I	8th	11th
		II	8th	11th
		III	11th	15th

* Treatment I (barley leaves with aphids always available); II (barley leaves with aphids only available two days per week); III (no barley leaves available).

## Data Availability

The data presented in this study are available on request from the corresponding author.

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
