# Peer review of "Effect of Abiotic Climatic Factors on the Gonadal Maturation of the Biocontrol Agent Sphaerophoria rueppellii (Wiedemann, 1830) (Diptera: Syrphidae)"

_insects, 2022, doi:10.3390/insects13070573_

Round 1
Reviewer 1 Report
The presented manuscript showed important information about the biology of one of the most important Biological control agents against aphids in crops. There is important new information, however the manuscript needs to be improved for publishing. Below some of my comments:
Tittle: I think is better to say under laboratory conditions instead environmental.
L40- I have the feeling that this sentence is too general, you can avoid starting the introduction like this and jump until L44
L49- I don't think this is necessary, perhaps its better to mention a short sentence about aphids but not too talk deeply about them, is not your aim
L51- I feel the need to connect this idea with a final sentence
L62- Can you mention in a small sentence how the temperature affects it?
L61- you can mention just the syrphid and avoid the other one, you said 2 spp and just mentioned 1, a little strange even if its not a syrphid spp.
L66-69- this paragraph needs to be summarized, is too long
L70- you can write this idea together with the previous paragraph of aphids.
L75- this need to be cited
I think you can add more bibliography about the morphological reproductive system, I need to read about it a little more, for other spp if there is nothing for your spp. I have the feeling that you made good work regarding the bibliography, but I think you can write the introduction in a better way, to connect better the ideas, now is a little like a puzzle.
L102- is better to start with your information, I mean, the material you used, the larvae which you used.
L120- can you explain more about the oviposition sites? Potted plants? Just leaves? With water?
L141- Described by
L165- which parameters? Can you describe it more deeply?
L174- this is not a result of this work, why did you put this reference?
In all the result section there is a lack of the statistical results
L237- “located”
L238- “abdomen”
L262- Is better to start with your results and not refer first to other Works.
L264- which temperature?
L272- This idea should be discussed more, you have nice results.
L278- “may - be”
L285- any discussion about this result?
L288- 297- this idea should be rewritten, you need to start with your results or to make a good connection, I have the feeling that you wrote more about the results of the references you cited and not of your results
L293-reference?
L301- and what does that mean? Please write more about this result
L305- you can mention the name of the author, is strange to read just the reference… trought all the MS
L307- you need to confirm your results with the results of your references and not the contrary.
L316- anything to mention about it? Maybe a suggestion? Or why do you think they have 2 dif ways of matting?---references
Author Response
A PDF with the answers to both reviewers is submitted.

Reviewer 2 Report
This article by José Orengo-Green and colleagues (insects-1762872) is well motivated, the structure is appropriate, and the manuscript is well written without missing any key details. The methods used are appropriate for the objectives of the work and, in general, well depicted. The resulting figures are sufficient, informative, and of good quality helping to follow the reasoning throughout the manuscript. The discussion of results and comments on future research was nicely done and will be useful to others. Overall, I enjoyed reading the manuscript. A few minor remarks have been made below for authors to consider.
Some of the authors statements would be much stronger if they tie their work to the body of literature that has built up on the bioecology and reproductive biology of mass-produced endo- and ectoparasite biocontrol agents (BCAs) for field releases. They all point to the same direction and should be paired back to this study. Some examples are J. Econ. Entomol. 112: 1560-1574 (mass produced ectoparasite BCAs) or J. Econ. Entomol. 112:1062-1072 (mass produced endoparasite BCAs), but there are others too. These studies provide strong evidence of increased longevity in BCAs reared at non-stressful low temperatures when compared to higher temperature regimes. This article should provide details on all these fronts to provide the proper context for the work, e.g., pages 303 through 316 in your text. They also indicate that the parasitism or egg load was significantly higher at intermediate temperatures (20-30C) than at cline margins (<15C or >35C), e.g., on lines 281 through 286 in your text. Adding these details will improve the paper in my opinion.
Good luck!
Author Response
A PDF with the response to both of the reviewers is submitted.

Round 2
Reviewer 1 Report
Dear authors,
Thank you for add the information I asked to you. The MS is much better and it has a better organization. I just have few more things, after that I complety agree about it publication in the journal. Congrats!:
Comment 15: L174- this is not a result of this work, why did you put this reference? Response: You are right, but we decided to do some trials to determinate the beginning of the oviposition day. For that reason, we decided to include it there. ▪ I think you need to mention this in the MS, in that way the reader will understand what you wrote to me as a response.
Comment 27: L305- you can mention the name of the author, is strange to read just the references and not the contrary. Response: We agree, but this is how it appears in the Journal instructions. ▪ Yes the instructions mention that, you need to write in that format, however you can write it in a nice way. Example (I copied this from a MS from the journal): which is parallel with the findings of HarunOr-Rashid [16] There are a lot of examples in the lastest manuscripts of the journal.
Comment 29: L316- anything to mention about it? Maybe a suggestion? Or why do you think they have 2 dif ways of mating?...references. Response: There is no information regarding to the ways of matting on Syrphidae We mentioned and posted the pictures because we consider it curious due to this character seems to have a phylogenetic significance. Perhaps is better to suppress it
I think is better since your aim was not this, you need more experiments to give results of this
Additional comments:
L188- “shows” or “showed”
L340- ¿?? What are these names? Maybe you forgot to put the reference in the correct way.
Author Response
A response to the reviewer comments is attached.
